# Transplantation of Chemical Compound-Induced Cells from Human Fibroblasts Improves Locomotor Recovery in a Spinal Cord Injury Rat Model

**DOI:** 10.3390/ijms241813853

**Published:** 2023-09-08

**Authors:** Toshihiro Kurahashi, Chiyoko Nishime, Eiko Nishinaka, Yuji Komaki, Fumiko Seki, Koji Urano, Yoshinori Harada, Toshikazu Yoshikawa, Ping Dai

**Affiliations:** 1Department of Cellular Regenerative Medicine, Graduate School of Medical Science, Kyoto Prefectural University of Medicine, 465 Kawaramachi-Hirokoji, Kamigyo-ku, Kyoto 602-8566, Japan; kurahasi@koto.kpu-ma.ac.jp (T.K.); yoshi@koto.kpu-m.ac.jp (T.Y.); 2Central Institute for Experimental Animals (CIEA), 3-25-12 Tonomachi, Kawasaki-ku, Kawasaki 210-0821, Japan; nishime@ciea.or.jp (C.N.); nishinaka@ciea.or.jp (E.N.); komaki@ciea.or.jp (Y.K.); fumiko@ciea.or.jp (F.S.); urano@ciea.or.jp (K.U.); 3Department of Pathology and Cell Regulation, Graduate School of Medical Science, Kyoto Prefectural University of Medicine, 465 Kawaramachi-Hirokoji, Kamigyo-ku, Kyoto 602-8566, Japan; yoharada@koto.kpu-m.ac.jp; 4Louis Pasteur Center for Medical Research, 103-5 Tanaka-Monzen-cho, Sakyo-ku, Kyoto 606-8225, Japan

**Keywords:** direct conversion, spinal cord injury, immature chemical compound-induced cells, chemical compounds, cell therapy

## Abstract

The development of regenerative medicine using cell therapy is eagerly awaited for diseases such as spinal cord injury (SCI), for which there has been no radical cure. We previously reported the direct conversion of human fibroblasts into neuronal-like cells using only chemical compounds; however, it is unclear whether chemical compound-induced neuronal-like (CiN) cells are clinically functional. In this study, we partially modified the method of inducing CiN cells (termed immature CiN cells) and examined their therapeutic efficacy, in a rat model of SCI, to investigate whether immature CiN cells are promising for clinical applications. Motor function recovery, after SCI, was assessed using the Basso, Beattie, and Bresnahan (BBB) test, as well as the CatWalk analysis. We found that locomotor recovery, after SCI in the immature CiN cell-transplanted group, was partially improved compared to that in the control group. Consistent with these results, magnetic resonance imaging (MRI) and histopathological analyses revealed that nerve recovery or preservation improved in the immature CiN cell-transplanted group. Furthermore, transcriptome analysis revealed that immature CiN cells highly express hepatocyte growth factor (HGF), which has recently been shown to be a promising therapeutic agent against SCI. Our findings suggest that immature CiN cells may provide an alternative strategy for the regenerative therapy of SCI.

## 1. Introduction

Spinal cord injury (SCI) is a neurological disease requiring no definitive care [1]. Surgical treatment and subsequent rehabilitation are the only types of care available for patients with SCI [2]. Depending on the degree and location of the SCI, recovery from neurodegeneration after SCI is limited. As neuronal cells in the central nervous system are difficult to regenerate once they are lost, cell replacement therapy is a promising strategy for treating SCI [3]. Various cell types have been studied as sources of cell replacement therapy for SCI treatment. Although some issues must be overcome, the use of neuronal stem cells/neuronal precursor cells (NSCs/NPCs), derived from embryonic stem cells (ESCs) and induced pluripotent stem cells (iPSCs), is promising [4,5]. Several studies using animal models of SCI have demonstrated that locomotion is recovered, after SCI, by the transplantation of NSCs/NPCs derived from ESCs and iPSCs [6,7]. However, several issues remain, including the time and cost involved in preparing the desired cells, tumorigenesis arising from genomic instability and stem cell contamination, as well as allogeneic transplantation [8,9].

Direct conversion (also known as direct reprogramming) methods have been developed to overcome problems associated with the use of stem cells for transplantation [10]. As this method converts a somatic cell into a different type of somatic cell without going through stem cells, the risk of tumorigenesis due to stem cell contamination is reduced. Terminally differentiated cells can be induced to undergo cell fate changes by the forced expression of a set of transcription factors (TFs), indicating that cell fate determination is reversible [11,12]. Fibroblasts can be directly converted into diverse functional cell types by the viral induction of known cell fate-determining TFs or microRNAs. By introducing pro-neural transcriptional factors or incorporating defined factors with either microRNAs or chemical compounds, the direct conversion of fibroblasts to neurons, termed induced neurons (iN), without going through the pluripotent state, has been established [13,14,15]. However, inducing conversion is complicated, and it requires a few months with low efficiency. In addition, the genomic integration of the viral vectors used for direct conversion has raised concerns regarding future applications of this approach.

Cell-permeable chemical compounds have been shown to promote cell reprogramming [16,17]. In addition, chemical compounds that modify critical signaling pathways are powerful tools to enhance, or even replace, defined genes [18,19]. iPSCs have been derived from mouse somatic cells by using chemical compounds or neural progenitor cells (iNPCs), generated using chemical cocktails under hypoxic conditions, without the introduction of exogenous genes [20,21]. Previous reports have also shown that chemical compound-induced neuronal-like (CiN) cells can be induced from human or mouse fibroblasts by using pure chemical cocktails without genetic manipulation [22,23,24]. Using our direct conversion method, human fibroblasts can be induced to differentiate into neuronal-like cells in approximately 3 weeks.

Here, we investigated whether immature CiN cells induced by a cocktail of chemical compounds, for only 8 d, had therapeutic effects in an SCI rat model by performing the Basso, Beattie, and Bresnahan (BBB) test and CatWalk analysis. Immature CiN cells were viable at the SCI site for at least 1 week after transplantation. In addition, although limited, the transplantation of immature CiN cells improved motor function recovery after SCI in a rat model. Furthermore, on MRI and histopathological analysis, we confirmed that nerve recovery or preservation tended to be improved and that there was decreased inflammation in the immature CiN cell-transplanted group compared to the control group. These data suggest that immature CiN cells have therapeutic potential in rat models of SCI.

## 2. Results

### 2.1. Immature CiN Cells Are Not Fully Differentiated into Neuronal-like Cells

Our previous study showed that a small chemical cocktail, i.e., the six compounds (6C) [TGF-β inhibitor SB-431542, bone morphogenic protein (BMP) inhibitor LDN-193189.

GSK-3β inhibitor CHIR99021, MEK-ERK inhibitor PD0325901, p53 inhibitor Pifithrin-α, and Forskolin] could induce CiN cells from human fibroblasts [24]. To further improve our induction method, we found that Dorsomorphin, BMP, and AMPK inhibitors accelerated the differentiation of CiN cells. When Dorsomorphin was added, the morphological changes in the induced cells were similar to those of 6C. Therefore, we decided to use Dorsomorphin to induce CiN cells by adding it to 6C in animal experiments.

Mature neurons are fragile, terminally differential, and non-plastic, making them unsuitable as a cell source for cell replacement therapy. On the other hand, NSCs/NPCs that can differentiate into various types of neurons and proliferate may be suitable for treating SCI [4,25]. In fact, numerous studies demonstrated the efficacy of cell therapy for SCI in animal models, by transplantation, with NSCs/NPCs derived from ESCs or iPSCs [26,27]. Based on the above evidence, we expected that our CiN cells might also have a therapeutic effect on SCI, by transplanting cells with cellular plasticity, in an immature state. Based on our preliminary results, we decided to use cells obtained 8 d after chemical compound treatment for transplantation in the SCI rat model (Figure 1A). Morphological changes in the cells seemed likely to be intermediate or immature. To qualitatively confirm the differentiation status of the cells, we conducted immunofluorescence studies to characterize CiN cells. After 8 d of treatment with the chemical cocktail, the cell morphology apparently changed, but the morphological changes seemed incomplete (Figure 1C, phase-contrast images). To investigate whether CiN cells became neuronal-like, we performed immunostaining for MAP2, which is a marker of mature neurons. As a result, the neuronal marker protein, MAP2, could not be clearly detected by immunostaining the cytoplasm of immature CiN cells (Figure 1C). In contrast, Tuj1, a neuronal marker, was weakly detected in immature CiN cells but not in control cells (Figure 1B,C). Furthermore, Synapsin I, a synaptic marker, was not detected in immature CiN cells. These results suggest that immature CiN cells did not differentiate into neuronal-like cells 8 days after the chemical compound induct.

### 2.2. Transplantation of the Immature CiN Cells Tends to Improve Functional Recovery in Subacute SCI Rat Model

We investigated the transplantability and therapeutic effects on SCI in a rat model of SCI. Immature CiN or control cells cultured in NM (see Section 4) were transplanted into rats 13 d after SCI. The SCI recovery status of the rats was observed for up to 6 weeks after transplantation. During the observation period, the body weights of the rats were not significantly different between the control cell-transplanted and immature CiN cell-transplanted groups (Appendix A).

In a pilot experiment, the rats were dissected 1 week after transplantation to confirm whether the transplanted cells were engrafted at the transplanted site. Immunohistochemical analysis was performed using an anti-STEM121 antibody, which can only detect proteins of human origin. STEM121-positive cells were detected at the transplantation sites (Appendix A). Although the engrafted STEM121-positive cells did not appear to extend neurites, they invaded the injured tissue, suggesting that they presented functional effects at the injury site, at least, in the first week after transplantation. The, 6 weeks after transplantation, STEM121-positive cells could not be found at the transplanted site.

To assess locomotor recovery in the SCI rat model transplanted with control cells or immature CiN cells, Basso, Beattie, and Bresnahan (BBB) tests were performed weekly before and after transplantation. Just 1 week after the SCI, the mean BBB scores for the groups transplanted with control cells and immature CiN cells were 2.3 ± 0.7 and 2.2 ± 0.3, compared with just 1 day before transplantation, the mean BBB scores for the groups transplanted with control cells and immature CiN cells were 3.6 ± 0.6 and 3.2 ± 0.7, respectively. There was no statistically significant difference between these values or between the two groups before transplantation. The mean BBB scores of the groups transplanted with control cells and immature CiN cells each week after SCI were as follows: 5.0 ± 0.2 and 5.1 ± 0.1 in the 3rd week, 5.1 ± 0.3 and 5.6 ± 0.3 in the 4th week, 5.8 ± 0.4 and 6.3 ± 0.3 in the 5th week, 5.8 ± 0.4 and 6.9 ± 0.4 in the 6th week, 5.9 ± 0.6 and 6.7 ± 0.5 in the 7th week, and 6.3 ± 0.6 and 6.9 ± 0.5 in the 8th week, respectively. Although a statistically significant difference was observed in the 6th week, after SCI, locomotor recovery tended to improve in the group transplanted with immature CiN cells throughout the observation period (Figure 2A). When evaluated in terms of the relative score plotted on the vertical axis, the mean relative score of the groups transplanted with control cells and immature CiN cells for each week was as follows: 1.4 ± 0.6 and 1.9 ± 0.7 in the 3rd week, 1.6 ± 0.5 and 2.3 ± 0.6 in the 4th week, 2.2 ± 0.4 and 3.1 ± 0.6 in the 5th week, 2.2 ± 0.6 and 3.7 ± 0.5 in the 6th week, 2.4 ± 0.6 and 3.4 ± 0.7 in the 7th week, and 2.7 ± 0.4 and 3.7 ± 0.6 in the 8th week, respectively. Statistically significant differences were also observed only in the 6th week after SCI (Figure 2B).

Furthermore, to validate the BBB test results, we performed a gait analysis using CatWalk XT. The mean of each analyzed dataset, in groups transplanted with control cells and immature CiN cells, was as follows: regularity index, 93.6 ± 2.1 and 97.3 ± 1.8 on day-13, 49.4 ± 8.4 and 43.4 ± 7.3 on day 47–49; stand, 0.19 ± 0.02 and 0.20 ± 0.02 on day-13, 0.26 ± 0.07 and 0.16 ± 0.03 on day 47–49; stride length, 9.15 ± 0.54 and 9.21 ± 0.38 on day-13, 5.58 ± 1.04 and 4.6 ± 0.78 on day 47–49; swing speed, 63.7 ± 5.3 and 66.9 ± 5.7 on day-13, 29.9 ± 4.1 and 48.9 ± 9.7 on day 47–49. The regularity index, stride length, and swing speed did not show a statistically significant difference, but stand, which means the total grounding time for a footprint, was significantly shorter in the immature CiN cell-transplanted group than in the control cell-transplanted group (Figure 2C–F). Collectively, the BBB test and CatWalk analysis results showed a partial, but significant, difference in locomotor recovery, after SCI, between the groups transplanted with control and immature CiN cells.

### 2.3. The Spinal Cord Injury Was Assessed by Performing MRI

Recent studies in primates, including humans, and rodents have revealed that MRI-based analysis is adequate for evaluation after SCI [28]. The therapeutic effect of immature CiN cell transplantation was observed in the SCI rat model, so we investigated what was happening at the injured site using MRI. The lesion epicenter, place of injury, 5 mm site on the rostral side (−5 mm), and 5 mm site on the caudal side (+5 mm) were analyzed using MRI (Figure 3A). By performing conventional MRI, axial T2-weighted images (T2WI, top panel), diffusion-weighted magnetic resonance imaging (DWI, middle panel), fractional anisotropy (FA, bottom panel), and apparent diffusion coefficient (ADC) were obtained (Figure 3B). Previous reports have shown that the FA of diffusion tensor imaging (DTI) correlates with motor function in human cervical spinal cord injury, whereas the ADC does not [29]. Therefore, to confirm whether the FA values obtained from the MRI analysis correlated with the BBB test results, we plotted these values and found strong correlations at all sites: −5 mm, the epicenter, and +5 mm (Figure 3C). Consistent with the BBB test and CatWalk analysis results, the FA values were not significantly different between the control and CiN cell-transplanted groups. The mean FA values of the groups transplanted with control cells and immature CiN cells were as follows: 0.30 ± 0.03 and 0.28 ± 0.02 at the −5 mm site, 0.37 ± 0.05 and 0.39 ± 0.03 at the epicenter, and 0.34 ± 0.03 and 0.42 ± 0.04 at the +5 mm site, respectively. The FA value at the +5 mm site showed the highest correlation with the BBB score. Although the difference was not statistically significant, the FA value in the CiN cell-transplanted group was higher than that in the control group at the +5 mm site (Figure 3D). The eigenvalues—λ1, λ2, and λ3—are calculated by performing a diagonalization transformation on the tensor obtained by the MRI analysis. FA values were calculated from these eigenvalues, so we examined each eigenvalue at the +5 mm site in detail. Eigenvalues were 92.2%, 85%, and 76.1% for λ1, λ2, and λ3 in the CiN cell-transplanted group compared to the controls (Appendix A). The results showed that λ3 was lower in the CiN cell-transplanted group than in the control group; however, the difference was not statistically significant.

### 2.4. Spinal Cord Area and Amounts of Myelin Sheath after SCI Tended to Be Larger at the Caudal +5 mm Site in the Immature CiN Cells Transplanted Group

Nerve preservation or regeneration after SCI correlate with the spinal cord area near the lesion epicenter [30], so HE staining was performed to compare the injured spinal cord area between the control and CiN cell-transplanted samples. The mean HE-stained area of the groups transplanted with control and immature CiN cells was as follows: 1.3 ± 0.13 and 1.4 ± 0.13 at the −5 mm site, 1.11 ± 0.11 and 1.02 ± 0.13 at the epicenter, 2.02 ± 0.22 and 2.39 ± 0.09 at the +5 mm site, respectively. Similar to the MRI results, although there was no statistically significant difference (Figure 4A,B), the immature CiN cell-transplanted group still had a larger spinal cord area than the control cell-transplanted group at the +5 mm site. The spinal cord area at the +5 mm site correlated well with the MRI FA values (R = 0.73, Figure 4C). MRI and HE staining showed a greater trend toward more axial fibers in the immature CiN cell-transplanted group, compared to the control group, only at the +5 mm site. Therefore, we focused on the +5 mm site to make a more detailed comparison between the control and immature CiN cell-transplanted groups. Luxol Fast Blue (LFB) staining was performed at the +5 mm site to evaluate the myelinated area (Figure 4D). The extent of the myelin sheaths correlated with the MRI FA values (R = 0.56, Figure 4E). The mean LFB positive area of the groups transplanted with control and immature CiN cells was 0.53 ± 0.11 and 0.61 ± 0.07, respectively (Figure 4F). The myelin sheath area was more prominent in the immature CiN cell-transplanted group than in the control cell-transplanted group. However, the difference was not statistically significant. These findings suggested that the immature CiN cell-transplanted group either suppressed nerve fiber atrophy or promoted regeneration at the +5 mm site.

### 2.5. Transplantation of the Immature CiN Cells Tends to Improve Motor Neuron Recovery and Reduce Inflammation in the Caudal +5 mm Site

Since the transplantation of immature CiN cells into the SCI rat model showed a trend toward motor function recovery, the number of motor neurons at the caudal +5 mm site, where there was a trend toward increased FA values, was examined by immunostaining with a choline acetyltransferase (ChAT) antibody. The results showed a trend toward slightly higher numbers of ChAT-positive cells in the immature CiN cell-transplanted group, 24.2 ± 1.5, than in the control cell-transplanted group, 22.1 ± 1.3. However, the difference was not statistically significant (Figure 5A,B).

Furthermore, because motor function recovery after SCI is related to the degree of inflammation, we examined the degree of inflammation at the caudal +5 mm site by immunostaining with an antibody against the ionized calcium-binding adapter molecule 1 (Iba1). As a result, there was a clear tendency for the Iba1-positive area to decrease in the immature CiN cell-transplanted group (11.7 ± 0.95%) compared to the control cell-transplanted group (15.1 ± 2.02%). However, this difference was not statistically significant (Figure 5C,D). This immunostaining result suggests that CiN cell transplantation suppresses inflammation at the caudal +5 mm site.

The motor neuron regeneration and motor function recovery after SCI are thought to be inhibited by glial scarring, which is referred to as reactive astrogliosis. It has been reported that reactive astrogliosis suppression promotes axonal regeneration and the recovery of function [31]. Therefore, we analyzed reactive astrogliosis using an antibody against the astrocytic marker, which is glial fibrillary acidic protein (GFAP). Consistent with the results showing a trend toward a higher number of ChAT-positive cells in the immature CiN cell-transplanted group, there was a decrease in the GFAP-positive area in the immature CiN cell-transplanted group (32.1 ± 0.73%) compared to the control cell-transplanted group (39.5 ± 6.81%) (Figure 5E,F). These results suggest that the transplantation of CiN cells may suppress reactive astrogenesis, contribute to motor neuron regeneration, and reduce inflammation.

### 2.6. Gene Expression of HGF and IL-1RN Was Enhanced in Immature CiN Cells

Why does immature CiN cell transplantation promote limited motor function recovery in the SCI rat model? To characterize immature CiN cells, we performed RNA-seq analysis to compare the gene expression profiles of control and immature CiN cells (Figure 6A). In immature CiN cells, 1083 genes were upregulated, and 1256 genes were downregulated compared to control cells (Figure 6A,B). As a result of the enrichment analysis of each upregulated and downregulated gene in the immature CiN cells, the expression of genes related to extracellular structural proteins was significantly upregulated in the immature CiN cells (Figure 6C). In contrast, the expression of genes related to viral responses was significantly downregulated in immature CiN cells (Figure 6C). Furthermore, when we compared the differential expression of cytokines, known to be associated with nerve regeneration after spinal cord injury, between the control and immature CiN cells, 30 genes were significantly upregulated in the immature CiN cells, and 34 genes were significantly downregulated (Figure 6D). Real-time RT-PCR was performed to confirm the RNA-seq results for several genes, and it was confirmed that the *HGF*, *BMP2*, and *IL-1RN* expressions were extremely high in immature CiN cells (Figure 6E). Particularly, HGF itself has recently undergone clinical trials for the treatment of SCIs [32], suggesting that it may play a significant role in the healing of SCIs. Therefore, we examined the extracellular secretion of HGF, at the protein level, using ELISA, and we found no detectable secretion in fibroblasts and a relatively small amount of secretion in control cells. In contrast, a significant amount of secretion was detected in immature CiN cells (Figure 7A). IL-1RN was also secreted by immature CiN cells, which was consistent with the gene expression results (Figure 7B). Although further analysis is required for other cytokines, Figure 7C shows a schematic diagram of the mechanism, underlying the therapeutic effect of immature CiN cells on SCI, predicted from gene expression.

### 2.7. Transplantation of CiN Cells Did Not Induce Tumorigenesis

To determine whether immature CiN cells were tumorigenic, we performed a tumorigenicity test using immunodeficient NOG mice. Immature CiN cells, prepared in the same manner as in the SCI experiment, were suspended in Matrigel at 3 × 10^5^ cells or 3 × 10^6^ cells, subcutaneously transplanted into NOG mice, and observed for 16 weeks. After observation, the transplanted sites were collected from euthanized mice and subjected to histopathological analyses (Figure 8A). Residual Matrigel was confirmed in some samples; however, tumor formation was not confirmed (Figure 8B). In addition, there was no significant difference in body weight between control mice injected with Matrigel, alone, and those transplanted with immature CiN cells throughout the observation period (Figure 8C). Furthermore, RNA-seq analysis showed that immature CiN cells exhibited lower gene expression levels of cancer-associated factors than fibroblasts and control cells (Figure 8D). These findings suggest that the transplantation of immature CiN cells provides extremely low tumorigenicity and does not cause concerns about tumor formation from immature CiN cell transplantation.

## 3. Discussion

Here, we showed that immature CiN cells, 8 d after chemical compound induction, expressed Tuj1, a neuronal marker. However, MAP2, a mature neuronal marker, may not differentiate into neuronal-like cells (Figure 1), although a noticeable morphological change was observed in these immature CiN cells (Figure 1C). We confirmed that the transplantation of immature CiN cells into the SCI rat model resulted in a limited recovery of motor function after SCI (Figure 2). MRI and histopathological analyses suggested that nerve preservation or regeneration possibly improved in the immature CiN cell-transplanted group compared to the control cell-transplanted group in the caudal +5 mm area from the lesion epicenter (Figure 3, Figure 4 and Figure 5). Consistent with these results, inflammation levels in the caudal +5 mm region tended to decrease in the immature CiN cell-transplanted group, although this was not statistically significant (Figure 5C,D). RNA-seq results revealed the elevated gene expression of various cytokines in immature CiN cells (Figure 6). HGF and IL-1RN were also secreted at the protein level (Figure 7). HGF has been suggested to be potentially effective in treating SCI [33,34], and clinical trials using HGF preparations are underway [32]. It is conceivable that HGF secreted by immature CiN cells exerts therapeutic effects in the SCI rat model.

The key to this study was the use of immature CiN cells. Neural cell therapy is preceded by the transplantation of neural tissues and terminally differentiated neurons. However, the discovery of NSCs/NPCs has become the starting point for transplantation research using these cells [35]. This trend of research suggests that immature cells are suitable for neural cell therapy. Furthermore, in recent years, research using NSCs/NPCs derived from ESCs/iPSCs has been active, particularly in applying iPSCs, which can use autologous cells to treat neurodegenerative diseases and have shown promise [25]. However, the use of pluripotent cells, including MSCs, raises concerns regarding tumor formation. To overcome these concerns regarding pluripotent cells, research on direct conversion has been conducted [10]. A growing body of research has focused on the direct conversion of cell fate using only small chemical compounds, which is a direct chemical conversion technique (also known as direct chemical reprogramming) [19,36]. Recently, it has been reported that even multipotent cells can be induced by small chemical compounds [37,38]. Therefore, developing a cell replacement therapy that uses a direct chemical conversion method is desirable. To the best of our knowledge, this is the first report of a confirmed therapeutic effect on small chemical compound-treated cells in an animal model of SCI.

A critical issue that needs to be resolved in the future is that, so far, we have not identified which immature CiN cells are more similar to existing cells. RNA-seq results revealed that gene expression of *HES1* and *SOX2*, which are NSC/NPC markers, is elevated in immature CiN cells compared to that in fibroblasts, while the gene expression of *NES* and *PAX6*, which are also NSC/NPC markers, is decreased (Appendix A). These data suggest that the immature CiN cells are not similar to NSCs/NPCs. Although the absence of tumorigenicity in immature CiN cells is indicated by the results of tumorigenicity tests and RNA-seq (Figure 8), a more detailed analysis of the properties and functions of immature CiN cells is needed in the future.

Another important question is: How are immature CiN cells metabolized in vivo following transplantation? Engraftment of immature CiN cells was confirmed up to 1 week after transplantation (Appendix A). However, we could not detect the engraftment of immature CiN cells near the graft site 6 weeks after transplantation. Presumably, the transplanted immature CiN cells become extinct sometime between one and six weeks after transplantation. Hence, it is unclear what type of cells the transplanted immature CiN cells will ultimately function in, and we need to trace immature CiN cells after transplantation, over time, in future studies to determine their whereabouts and fate. Immature CiN cells are ideal for cellular medicine if they can exert a therapeutic effect after transplantation and then disappear without adversely affecting the surrounding environment.

We should also consider the timing of the immature CiN cell transplantation, as HGF has been reported to be effective in the early stages of SCI [39], and early transplantation of immature CiN cells may be desirable. On the other hand, realistically, more detailed studies are needed to determine how far the induction period can be shortened to use autologous cells for treatment after SCI. However, in this study, the therapeutic effect was assessed 13 d after SCI, which is a subacute stage. This indicates that cytokines, other than HGF, secreted by immature CiN cells are also important for therapeutic effects in the SCI rat model.

As shown in Panel C of Figure 7, the gene expression of cytokines, associated with anti-inflammatory responses after SCI, was elevated in immature CiN cells. IL-1RN and CXCL12 (also known as SDF-1) are involved in the anti-inflammatory response following SCI [40,41]. In the immature CiN cell-transplanted group, there was a trend toward a decrease in the GFAP-positive area in the spinal cord +5 mm from the lesion site to the dorsal side 6 weeks after transplantation (Figure 5E,F). However, TNFAIP6 (also known as TSG-6) and TGF-β2 are involved in glial scar formation and anti-inflammatory responses, making it difficult to interpret their roles in the healing effects of immature CiN cells in SCI [42,43]. As the details of the effects of the inflammatory response on injury and recovery after SCI are still unclear, further research in this area is warranted [44]. The increased expression of BDNF, HGF, BMP2, and BMP4, which are involved in neurogenesis in immature CiN cells, may contribute to the therapeutic effect of immature CiN cell transplantation in a rat model of SCI [45,46]. As HGF promotes neurogenesis after SCI and decreases astrocytic scar formation [47], the decreased percentage of GFAP-positive areas in the immature CiN cell transplantation group may be due to HGF function. Based on the above, we speculate that transplanted immature CiN cells may contribute to the recovery of motor function in the SCI rat model by providing paracrine factors as bystanders.

If cells that can control the levels of HGF and other cytokines in immature CiN cells can be created, cellular medicine will be more effective in treating SCI. Alternatively, in vivo reprogramming, using small-molecule chemical compounds, may be clinically applicable [48]. The clinical application of immature CiN cells is also promising because HGF may help treat many other neurodegenerative diseases and spinal disorders [49]. In addition, although many transplantation experiments have examined the therapeutic effect in a subacute-stage SCI model, the therapeutic effect in a chronic SCI model needs to be verified.

Our experimental results suggest that this direct chemical conversion strategy is a potential cell replacement therapy for neurological disorders and in regenerative medicine. Furthermore, because the pathophysiology of SCI is complex and its pathological mechanisms are not fully understood [50], combining this cell therapy with other technologies, such as biomaterials, may lead to a higher therapeutic effect.

## 4. Materials and Methods

### 4.1. Cell Culture and Induction of Immature CiN Cells

Human fibroblasts were purchased from LIFELINE CELL TECHNOLOGY (FC-0024, Frederick, MD, USA) or Zenbio Inc. (CC-2511, Durham, NC, USA). Cells were seeded at 1.0 × 10^5^ cells in a 35 mm dish in DMEM high-glucose medium (11995-065, Thermo Fisher Scientific, Waltham, MA, USA) supplemented with 10% FBS (SH30088, HyClone, Cytiva, Marlborough, MA, USA), 100 U/mL penicillin, and 100 μg/mL streptomycin (15140122, Thermo Fisher Scientific, Waltham, MA, USA). To generate immature CiN cells, on day 3, the culture medium was switched from DMEM to neuronal medium (NM) composed of advanced DMEM/F12 [1% (*v*/*v*) N2 supplement, Gibco] and neurobasal [2% (*v*/*v*) B27 supplement, Gibco, Thermo Fisher Scientific, Waltham, MA, USA] mixed at a 1:1 ratio and cultured for 8 d. When indicated, small chemical compounds were added to NM to reach the following final concentrations: SB-431542 (2 μM, Wako, Osaka, Japan), LDN-193189 (1 μM, Wako, Osaka, Japan), CHIR99021 (1 μM, Wako, Osaka, Japan), PD0325901 (1 μM, Wako, Osaka, Japan), Pifithrin-α (5 μM, Wako, Osaka, Japan), Forskolin (7.5 μM, Wako, Osaka, Japan), and Dorsomorphin (1 μM, Wako, Osaka, Japan).

### 4.2. Immunocytochemical Analysis

Immunocytochemical analyses were performed according to a standard protocol. Briefly, cells were fixed in 2% paraformaldehyde (Nacalai Tesque, Kyoto, Japan) for 1 h at room temperature, washed 3 times with PBS, and incubated in PBS containing 0.1% Triton X-100 for 10 min, as well as in 3% skim milk/PBS for 2 h at room temperature. The fixed cells were then incubated with the following primary antibodies diluted in 3% skim milk/PBS at 4 °C overnight: mouse anti-III-tubulin (MMS-435P, 1:2000, BioLegend, San Diego, CA, USA) or rabbit anti-MAP2 (AB5622, 1:1000, MilliporeSigma, Burlington, MA, USA). After washing three times with PBS, the cells were incubated with secondary antibodies diluted in 3% skim milk/PBS: Alexa Fluor 488 goat anti-mouse IgG (1:1000, Thermo Fisher Scientific, Waltham, MA, USA) or Alexa Fluor 594 goat anti-rabbit IgG (1:1000, Thermo Fisher Scientific, Waltham, MA, USA) for 2 h at room temperature. The DNA was stained with DAPI (D523, 1:1000; DOJINDO, Kumamoto, Japan) for 15 min. Images were analyzed using a fluorescence microscope (Axio Vert.A1,Carl Zeiss, Oberkochen, Germany).

### 4.3. Animals

For the animal model of spinal cord injury (SCI), healthy female F344/NJcl-rnu/rnu (Athymic Nude) rats (6 weeks old, 120–140 g) were purchased from CLEA Japan, Inc. (Tokyo, Japan). For the tumorigenicity test, and male NOD/Shi-*scid* IL2rg^null^ (NOG) mice were purchased from CLEA Japan, Inc. (Tokyo, Japan). The rats and mice had ad libitum access to standard chow and tap water. The animal room was maintained under specific pathogen-free conditions, at a constant temperature of 20–22 °C, with a 12 h alternating light–dark cycle. All animal experiments were performed per institutional guidelines, and we obtained the animal ethics committee's approval. The approval number is 21072A for the SCI experiment and M2021-559 for the tumorigenicity test, which were approved by the Institutional Animal Care and Use Committee of the CIEA and the Ethics Committees of Kyoto Prefectural University of Medicine, respectively.

### 4.4. Contusive Spinal Cord Injury in Rats

Athymic rats were pretreated with an intraperitoneal (i.p.) ampicillin injection (50 mg/kg) and a hypodermic injection of 4 mL saline, and then, they were anesthetized with an i.p. injection of three types of mixed anesthetic agents, comprising medetomidine hydrochloride (0.375 mg/kg), midazolam (2 mg/kg), and butorphanol (2.5 mg/kg). After laminectomy at the level of the tenth thoracic vertebra (T10), the lamina of T10 was exposed, followed by spinal cord impact at a force of 220 kdyn using an Infinite Horizon impactor (IH-400,Precision Systems and Instrumentation, Lexington, KY, USA). After surgery, the rats were administered Atipamezole (0.75 mg/kg) and Buprenorphine (0.05 mg/kg) and placed on a temperature-controlled mat (37–38 °C) until awakening. The animals received Ampicillin, Buprenorphine, and saline daily, for 3 days following surgery, to prevent postoperative infections.

### 4.5. Preparation of Cells and Transplantation

For transplantation into the SCI rat model, human fibroblasts were seeded at 2.25 × 10^6^ cells in a 225 cm^2^ flask (431082, CORNING, Corning, NY, USA). After 8 days of culture with (immature CiN cells) or without (control cells) the 7 chemical compounds mentioned in the main text, the cells were harvested and resuspended in NM. Then, 1 day before transplantation, body weight measurements and BBB tests were performed. The rats were divided into two groups: immature CiN cells and control cells. To make the variance of the body weight and the BBB score uniform between the two groups, the grouping software “Simple Grouping ver. 1.0.5” (H&T Co., Osaka, Japan) was used. Additionally, 13 days after SCI (subacute stage of SCI), dissociated immature CiN cells or control cells (approximately 0.6 × 10^6^ cells/5 μL in NM medium without chemical compounds) were injected into the injured site, at a rate of 1 μL/min, using a 26-gauge steel needle attached to a Hamilton syringe and a stereotaxic microinjector (Legato 130, KD Scientific, Holliston, MA, USA).

### 4.6. Basso, Beattie, and Bresnahan Test

Locomotor recovery after SCI was assessed using the Basso, Beattie, and Bresnahan (BBB) open field test [51]. Therefore, 2 independent researchers scored (0–21) the locomotor ability of injured rats in a circular arena for 5 min every week for 6 weeks after transplantation and just 1 d before transplantation for the grouping of the rats.

### 4.7. CatWalk Analysis

The CatWalk system (CatWalk XT, Noldus Information Technology, Wageningen, The Netherlands) comprises an objective test to detect functional changes in the SCI model. Automated gait analysis was performed before and by the CatWalk System equipped with a high-speed color camera and software (CatWalk XT version: 10.6.608). The CatWalk system has been previously described [52]. Rats crossed an illuminated walkway (glass plate) to collect paw prints. The data for completing the three steps were considered successful. Compliant gait was classified for all limbs and analyzed statistically.

### 4.8. Magnetic Resonance Imaging (MRI)

Rats were anesthetized with isoflurane (VTRS, Viatris, Canonsburg, PA, USA), and their respiratory rates and body temperatures were stabilized before performing the MRI measurements. The measurements were carried out using a 7.0 tesla MRI system, equipped with actively shielded gradients, at a maximum strength of 700 mT/m (Biospec 70/16, Bruker BioSpin, Ettlingen, Germany), a 72 mm volume transmit coil (Bruker BioSpin, Ettlingen, Germany), and a custom-made 4 ch surface coil (Takashima seisakusho, Tokyo, Japan).

The DTI measurements were performed as previously described [53]. The anatomical sagittal section was acquired by using a rapid acquisition with a relaxation enhancement (RARE) sequence with the following parameters: TE of 50 ms, TR of 2000 ms, the RARE factor of 16, 12 averages, FOV of 51.2 × 38.4 mm, matrix of 256 × 192, slice thickness of 1.2 mm, five slices, and a scan time of 4 min. The axial section was acquired using a RARE sequence with the following parameters: TE of 50 ms, TR of 2300 ms, the RARE factor of 16, 12 averages, FOV of 38.4 × 38.4 mm, matrix of 256 × 256, slice thickness of 0.9 mm, 17 slices, and a scan time of 5 min and 59 s. DTI measurements were performed using a Spin-echo Diffusion MRI with the following parameters: TE 16 ms, TR 1500 ms, 1 average, b-value of 1000 s/mm^2^, 12 directions, FOV of 38.4 × 38.4 mm, matrix 128 × 128, slice thickness of 0.9 mm, 17 slices (axial), and scan time of 20 min 48 s.

The DTI images were processed using the diffusion toolkit [54] to calculate the diffusion-weighted image (DWI), fractional anisotropy (FA), and apparent diffusion coefficient (ADC). In addition, regions of interest (ROIs) were defined in the spinal cord substance, without cerebrospinal fluid (CSF), in the epicenter, rostral 5 mm (−5 mm), and caudal 5 mm (+5 mm) slices, and FA and ADC were measured.

### 4.9. Histopathological Analysis

Following whole-body perfusion with PBS and 4% paraformaldehyde phosphate buffer solution (Nacalai Tesque, Kyoto, Japan), the excised spines containing the spinal cords were decalcified. Spinal cords with decalcified spines were resected at a thickness of 5 mm and embedded in paraffin. Subsequently, slices were obtained from paraffin-embedded specimens at 4 μm thickness. Finally, the standard method subjected the serial sections to be stained with hematoxylin and eosin (H&E) staining and Luxol fast blue (LFB) staining. Sections were scanned using a semi-automated digital microscope (NanoZoomer 2.0-RS, Hamamatsu Corporation, Shizuoka, Japan) and reviewed using a Nanozoomer Digital Pathology viewer (NDP.view, version 2, Hamamatsu Corporation, Shizuoka, Japan). The exported images were analyzed using Fiji software (ImageJ ver. 1.54f, National Institute of Health, Bethesda, MD, USA). The images were divided into channels and converted into binary images using a set threshold value. The HE or LFB-positive spinal cord areas were quantified automatically using the Analyzed Particle tool.

### 4.10. Immunohistochemical Analysis

Serial sections, obtained in the same manner as for the histopathological analysis, were subjected to histopathological analysis. Deparaffinized sections were incubated with the following primary antibodies: a mouse monoclonal antibody specific for the human cytoplasmic marker STEM121 (1:1000, Takara Bio Inc., Shiga, Japan), the motor neuron marker choline acetyltransferase (ChAT, AB144P, MerckMillipore, Burlington, MA, USA), the macrophage marker Iba1 (011-27991, Wako, Osaka, Japan), and the astrogliosis marker GFAP (G3893, Sigma-Aldrich, St. Louis, MO, USA). Signals were visualized using a horseradish peroxidase-conjugated goat anti-mouse antibody with diaminobenzidine. The quantification of Iba1 and GFAP-positive cells was performed using the Fiji software.

### 4.11. RNA-Seq Analysis

Total RNA was isolated and purified from three biological replicates of control cells (Cont) and immature CiN cells by using the FastGene RNA Premium Kit (Nippon Genetics, Tokyo, Japan). cDNA library creation and RNA-seq were performed by Macrogen Japan Co. (Tokyo, Japan). A cDNA library was constructed using the TruSeq Stranded mRNA LT Sample Prep Kit (Illumina, San Diego, CA, USA), according to the manufacturer’s protocol. The resulting 101 bp paired-end cDNA libraries were sequenced using a NovaSeq 6000 (Illumina, San Diego, CA, USA). Trimmed reads were mapped to a reference genome (NCBI GRCh38) using HISAT2. The known genes and transcripts were assembled by using StrigTie based on a reference genome model. The expression profile was calculated for each sample and transcript/gene as read counts and FPKM (fragments per kilobase of transcript per million mapped reads). Differentially Expressed Genes (DEGs) analysis was analyzed using edgeR. The results satisfied |fc| ≥ 2 and an exact test raw *p*-value < 0.05 conditions. The DEGs list was further analyzed by using the g:Profiler tool for gene set enrichment and gene ontology analysis. Heat maps were generated using Heatmapper.

### 4.12. Real-Time RT-PCR

Total RNA was extracted from the cultured cells and purified using a FastGene RNA Basic Kit (FG-80250, Nippon Genetics, Tokyo, Japan). cDNA was synthesized using the ReverTra Ace qPCR RT Master Mix with gDNA Remover (FSQ-301, TOYOBO, Osaka, Japan). Real-time PCR analysis was performed using the QuantStudio 3 real-time PCR system with Power SYBR Green PCR Master Mix (4367659, Thermo Fisher Scientific, Waltham, MA, USA) per the manufacturer’s recommendations. Relative mRNA expression levels were calculated using the delta–delta Ct method. *TBP* was used as the internal control. The primer sequences are listed in Appendix A.

### 4.13. Enzyme-Linked Immunosorbent Assay (ELISA)

The culture medium was replaced with fresh medium 6 days after the start of induction, and the culture supernatant was collected 48 h later and subjected to ELISA. The ELISA was performed using AuthentiKine™ Human HGF ELISA Kit (PGI-KE00168-1, Proteintech, Rosemont, IL, USA) for HGF and Human IL-1ra/IL-1F3 Quantikine ELISA Kit (DRA00B, R&D SYSTEMS, Minneapolis, MN, USA) for IL-1RN, according to the manufacturer’s instructions, respectively.

### 4.14. Tumorigenicity Test

NOG mice were anesthetized by an intraperitoneal injection of three types of mixed anesthetic agents: medetomidine hydrochloride (0.375 mg/kg), midazolam (2 mg/kg), and butorphanol (2.5 mg/kg). After depilating the hair of the shoulder region of the mice, cells suspended in Matrigel were subcutaneously transplanted into both shoulders of the mice at the following numbers: 1 × 10^3^ HeLa cells as a positive control and 3 × 10^5^ or 3 × 10^6^ immature CiN cells. Matrigel was used as the negative control.

The observation was continued for 16 weeks after transplantation. In the event of abnormalities, such as a tumor diameter of over 17 mm, a volume of over 1/10th of the body weight, or rapid weight loss of over 20%, the mice were euthanized by inducing an overdose of barbiturate anesthetics as the humane endpoint. At the end of the observation period, the mice were euthanized, without pain, using an overdose of barbiturate anesthetic, and the transplanted site was removed and fixed, followed by various histopathological evaluations.

### 4.15. Statistical Analysis

All results are expressed as the mean ± SEM. GraphPad Prism software ver. 4 was used for statistical analyses. Single comparisons between two groups were performed using an unpaired two-tailed Student’s *t*-test. BBB scores were analyzed using the Mann–Whitney U-test. Multiple comparisons were performed using the one-way ANOVA, followed by the Tukey–Kramer test. A *p*-value of less than 0.05 was considered significant. *; *p* < 0.05, **; *p* < 0.01, ***; *p* < 0.001., n.s.; not significant.

## Figures and Tables

**Figure 1 ijms-24-13853-f001:**
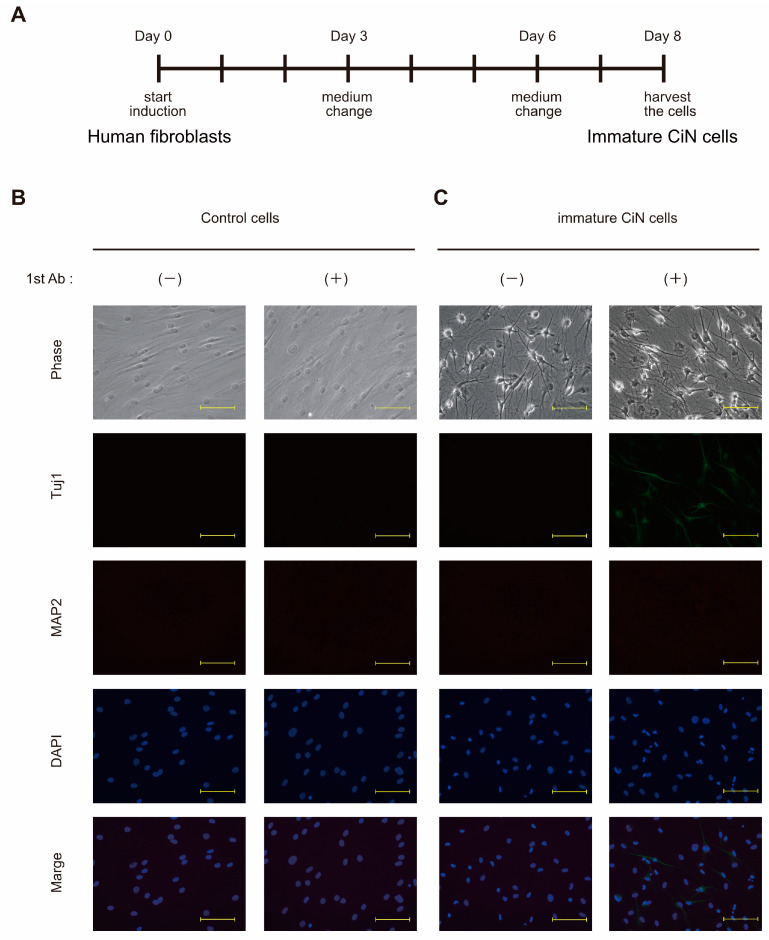
Immature CiN cells are not yet neuronal-like cells. (**A**) A schematic representation of the induction protocol for immature CiN cells. (**B**) Control cells cultured for 8 d without 7 chemical compounds were stained with anti-Tuj1 (green) and anti-MAP2 (red) antibodies, which are neural cell markers. (**C**) Immature CiN cells were cultured for 8 d with 7 chemical compounds and stained with anti-Tuj1 and anti-MAP2 antibodies, which are neural cell markers. Nuclei were stained with DAPI (blue): scale bar, 200 μm.

**Figure 2 ijms-24-13853-f002:**
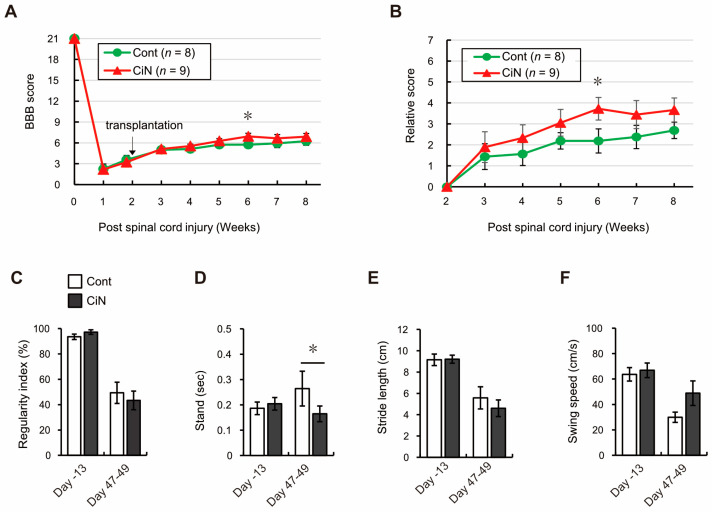
The transplantation of CiN cells promoted motor recovery in a limited way in the SCI rat model. Hindlimb motor function was measured by BBB test every week up to 6 weeks after transplantation. Panel (**A**) shows measured values, and panel (**B**) shows relative values (calculated by considering the score just before transplantation as one). Hind paws (**C**) regularity index, (**D**) stand, (**E**) stride length, and (**F**) swing speed were measured by CatWalk XT just before (Day-13) and 47–49 days after SCI (*n* = 4 animals in the control cell-transplanted group and *n* = 4 animals in the CiN cell-transplanted group). White and gray columns represent the control and the immature CiN cell-transplanted group, respectively. * *p* < 0.05.

**Figure 3 ijms-24-13853-f003:**
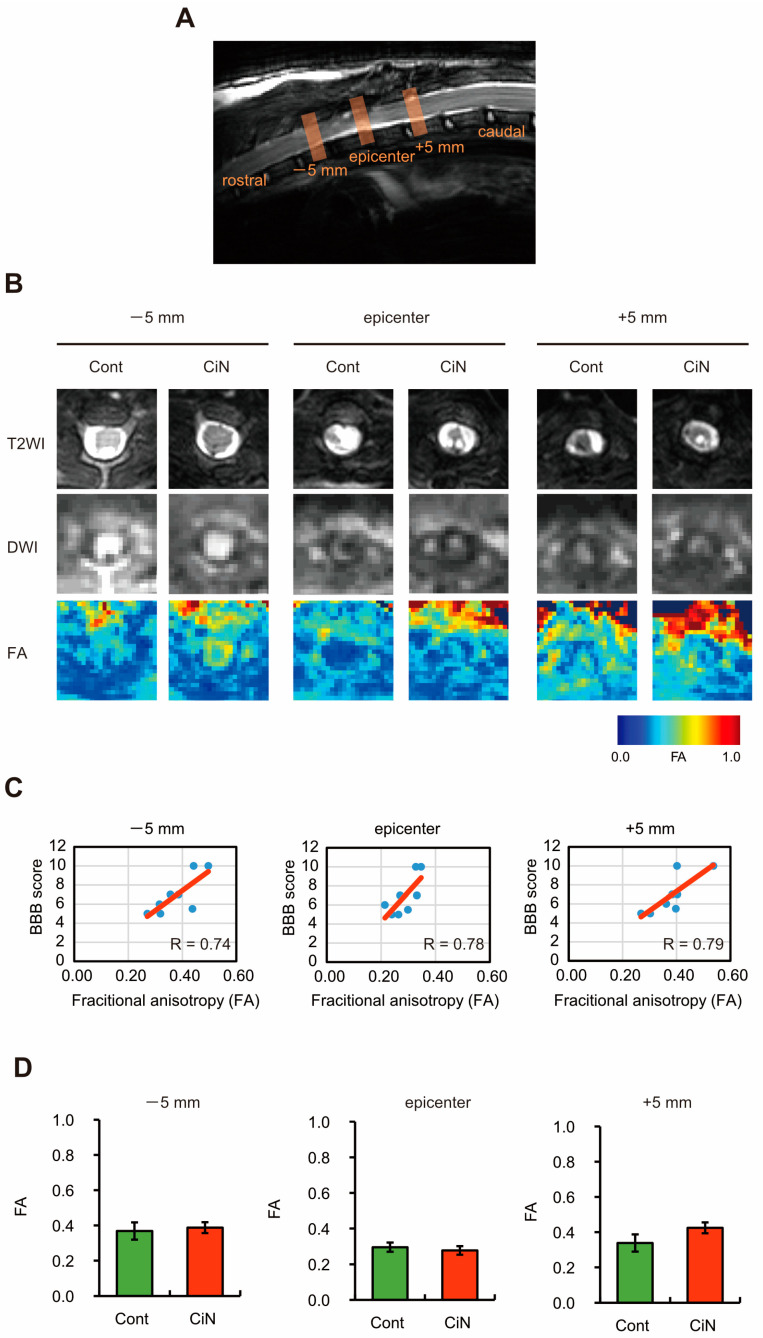
Magnetic resonance imaging (MRI) analysis of the injured spinal cord. (**A**) Sagittal T2 weighted MR imaging of the spinal cord around the lesion epicenter site 8 weeks after SCI. (**B**) Axial T2 weighted MR imaging (T2WI), diffusion-weighted MR imaging (DWI), and fractional anisotropy (FA) of the injured spinal cord around the lesion epicenter site 8 weeks after SCI. The image of the data closest to the mean is shown. (**C**) A scatter plot showing the relationship between the BBB score and FA value at the rostral (−5 mm; left panel), epicenter (center panel), and caudal (+5 mm; right panel) sites. (**D**) Each FA value in panel B is graphed (*n* = 4 animals in the control cell-transplanted group and *n* = 4 animals in the CiN cell-transplanted group). The data are expressed as group means ± SEM. Abbreviations: SCI, spinal cord injury; BBB test, Basso, Beattie, and Bresnahan test; cont, control.

**Figure 4 ijms-24-13853-f004:**
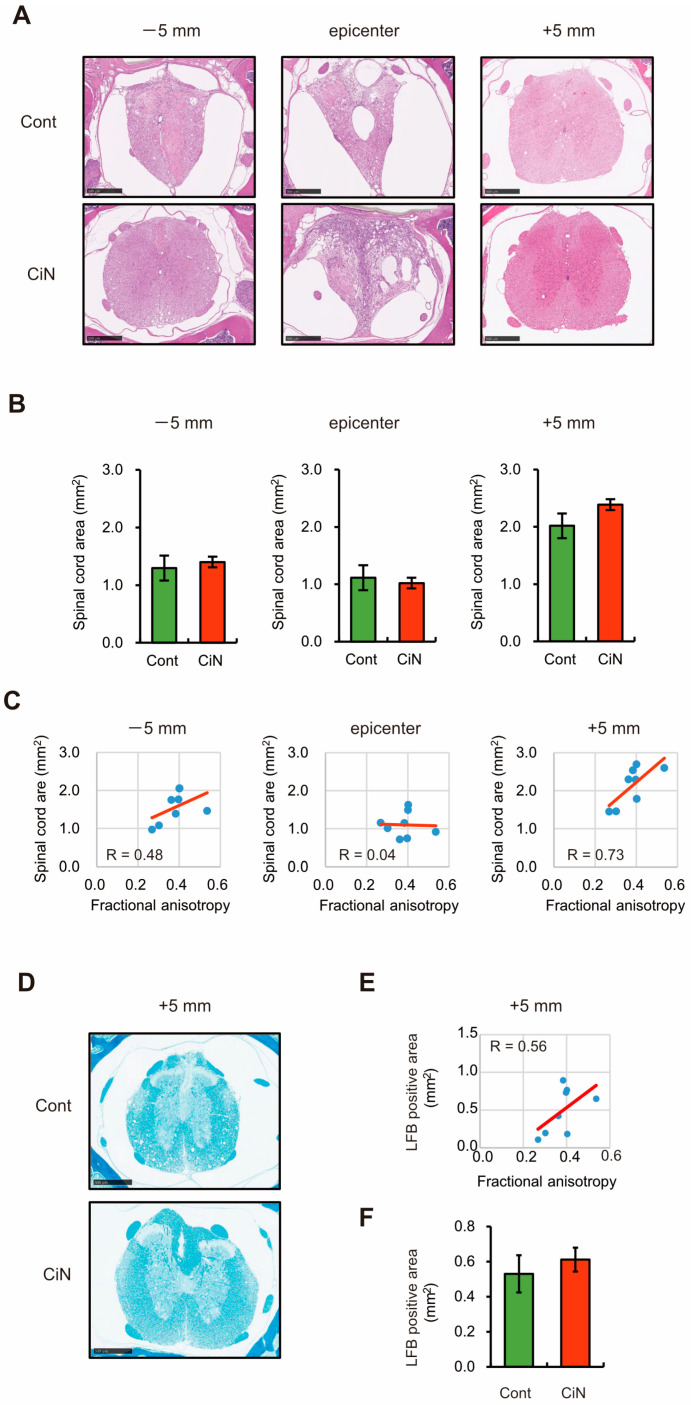
Histopathological analysis of the spinal cord around the lesion site. (**A**) HE-staining of coronal sections at the rostral (−5 mm), epicenter, and caudal (+5 mm) segments after SCI. The image shown includes the data closest to the mean. Scale bar, 500 μm. (**B**) Quantitative analysis of the HE-stained area in the axial sections (*n* = 8 animals in the control cell-transplanted group and *n* = 9 animals in the CiN cell-transplanted group). (**C**) A scatter plot shows the relationship between the spinal cord area (HE-stained area) and FA value at the rostral (−5 mm; left panel), epicenter (center panel), and caudal (+5 mm; right panel) sites. (**D**) LFB-staining of coronal sections at caudal (+5 mm) segments after SCI. The image shown includes the data closest to the mean. Scale bar, 500 μm. (**E**) Quantitative analysis of the LFB-stained area in the axial sections (*n* = 8 animals in the control cell-transplanted group and *n* = 9 animals in the CiN cell-transplanted group). (**F**) A scatter plot shows the relationship between the LFB-stained area and FA value at the caudal (+5 mm) site.

**Figure 5 ijms-24-13853-f005:**
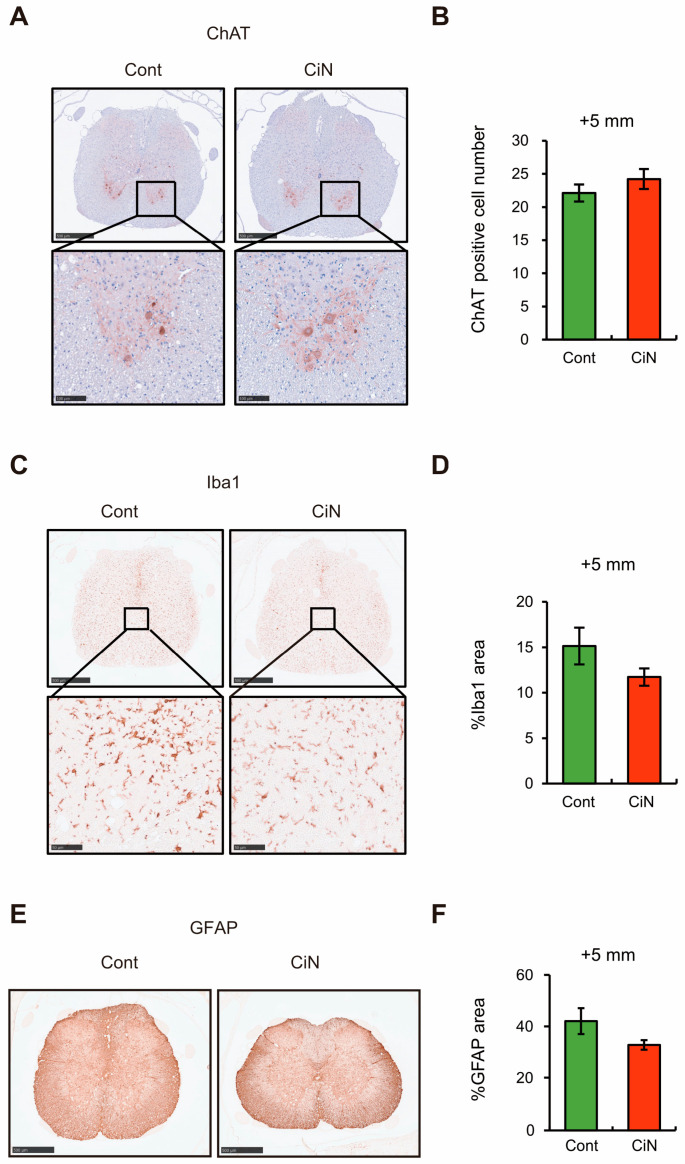
Immunohistochemical analysis of the spinal cord around the lesion site. (**A**) Immunostaining of ChAT-positive motor neurons in the spinal cord at the caudal site from the lesion epicenter. Scale bar, 500 μm. The bottom panels are areas magnified from the upper panels. Scale bar, 100 μm. The image shown includes the data closest to the mean. (**B**) The number of ChAT-positive motor neurons per section. (**C**) Immunostaining of Iba1 positive microglia/macrophages in the spinal cord at the caudal site from the epicenter. Scale bar, 500 μm. The bottom panels are magnified areas from the upper panels. Scale bar, 100 μm. The image shown includes the data closest to the mean. (**D**) The results were quantified as the percentage (%) of Iba1-positive area to spinal cord area, as calculated by HE-staining. (**E**) Immunostaining of GFAP-positive astrocytes in the spinal cord at the 5 mm caudal site from the lesion epicenter. The image shown includes the data closest to the mean. Scale bar, 500 μm. (**F**) The results were quantified as the percentage (%) of GFAP-positive area to spinal cord area, as calculated by HE-staining. All experiments were performed on animals with *n* = 8 in the control cell-transplanted group and *n* = 9 in the immature CiN cell-transplanted group.

**Figure 6 ijms-24-13853-f006:**
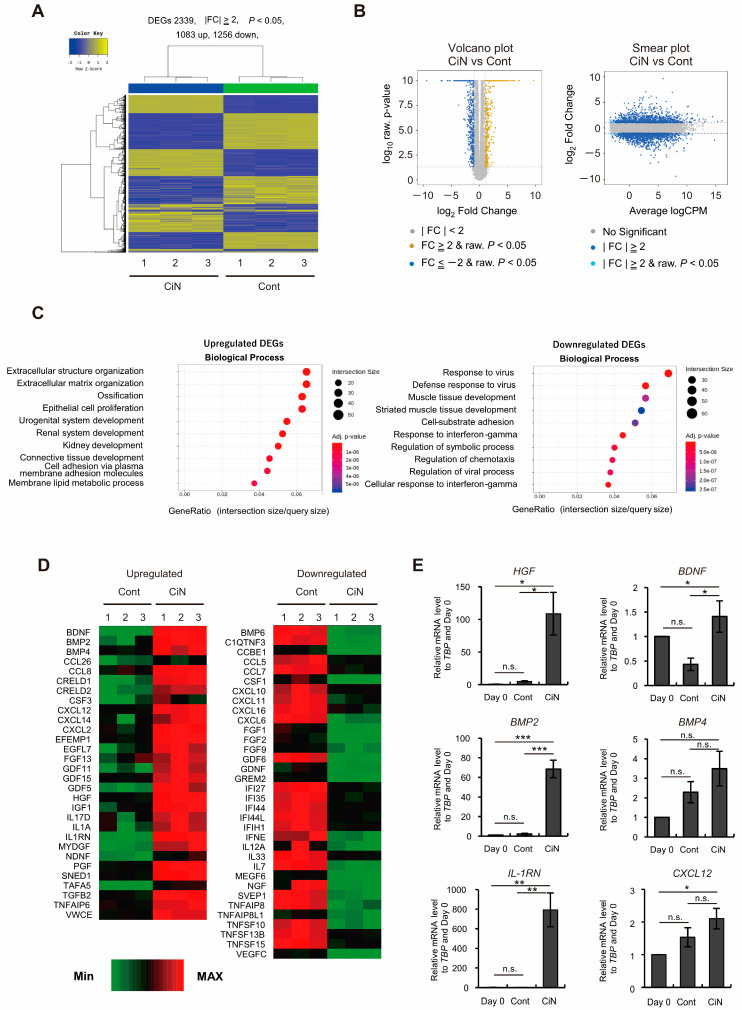
Transcriptome analysis for immature CiN cell characterization. (**A**) The heat map shows hierarchical clustering analysis, which represents 2339 differentially expressed genes (DEGs) (|fold change (FC)|≥ 2, *p* < 0.05) between control and immature CiN cells. (**B**) Volcano and Smear plots indicate logarithmic FC (fold change), *p*-value, and CPM (counts per million) between control and immature CiN cells. (**C**) The Enrichment analysis, based on the Gene ontology (GO) database, was conducted, with a significant gene list, in the upregulated and downregulated DEGs. The top 10 GO terms are represented in the categories of biological processes. (**D**) Transcriptional profiles are shown as heat maps in functional groups of cytokines. The color scale shows z-scored fragments per kilobase of transcript per million mapped sequence reads (FPKM), thereby representing mRNA levels of each gene in green (lower expression) and red (higher expression). (**E**) Several upregulated cytokine genes were confirmed by real-time RT-PCR analysis. A *p*-value of less than 0.05 was considered significant. *; *p* < 0.05, **; *p* < 0.01, ***; *p* < 0.001., n.s.; not significant.

**Figure 7 ijms-24-13853-f007:**
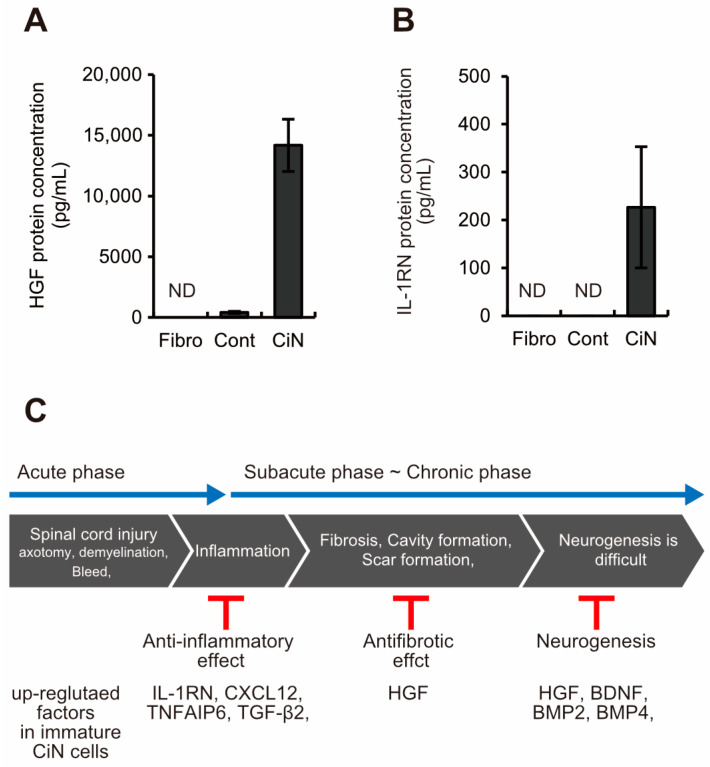
Immature CiN cells secrete HGF and IL-1RN at the protein level. (**A**,**B**) ELISA analysis revealed that CiN cells secrete HGF and IL-1RN, whereas fibroblasts and control cells do not. (**C**) A schematic diagram of cytokines with elevated gene expression in immature CiN cells and their expected effects on SCI.

**Figure 8 ijms-24-13853-f008:**
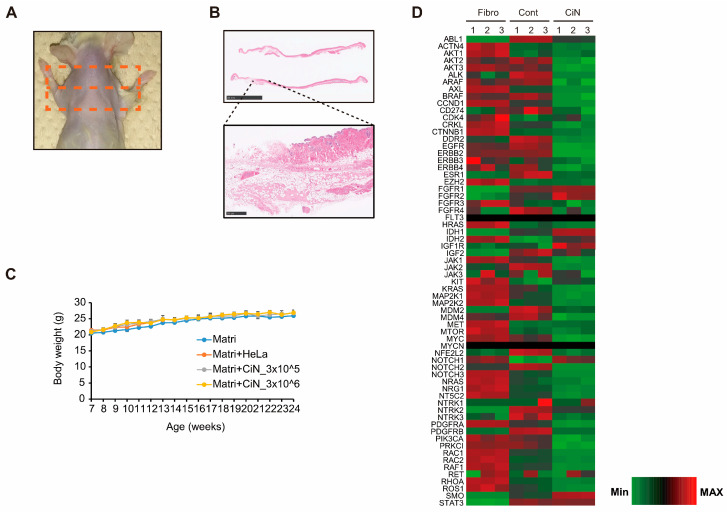
Immature CiN cells are almost certainly not tumorigenic. (**A**) After euthanasia, the shoulder of the mouse (where the cells were implanted) was shaved. Skin from the area, circled by the orange square, was collected for histopathological analysis. (**B**) HE-staining images of skin samples from panel (**A**). Scale bar, 10 mm. The bottom panels are magnified areas from the upper panels. Scale bar, 500 μm. (**C**) Body weight change in each group from the time of transplantation to the end of observation (*n* = 3). (**D**) Transcriptional profiles are shown as heat maps in functional groups of tumorigenesis-related genes. The color scale shows z-scored fragments per kilobase of transcript per million mapped sequence reads (FPKM), representing the mRNA levels of each gene in green (lower expression) and red (higher expression).

## Data Availability

Not applicable.

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
