# Peer review of "Transplantation of Chemical Compound-Induced Cells from Human Fibroblasts Improves Locomotor Recovery in a Spinal Cord Injury Rat Model"

_ijms, 2023, doi:10.3390/ijms241813853_

Round 1

Reviewer 1 Report

This study by Kurahashi and colleagues proposes the use of a human fibroblast-derived, chemically induced, immature neuronal-like cell type for transplantation in a rat model of spinal cord injury. To date, the number and variety of experimental studies employing cell grafts are very large, but a successful cell replacement approach with anatomical and functional graft integration remains a challenge. A lot of cell models have been used, including fetal neurons, adult neuronal stem/progenitor cells, mesenchymal stem cells from different origins, neural cells obtained from embryonic stem cells or from iPSCs cells, and so on, and variable effects, sometimes promising, have been demonstrated, mainly depending on trophic support provided by the grafts. In my opinion, the present manuscript represents an additive work in the field, but the scientific soundness and the translational perspectives are quite limited, for the following reasons:

1)      Authors apparently tried to include this work among those using a cell-replacement strategy, but they did not show any cell that is derived from the graft.

2)      In particular, the graft has not been adequately studied, as just a pilot experiment has been made to evaluate engrafting only at one week after transplantation. Since transplanted animals have been evaluated for eight weeks after transplantation, what is the reason for not having evaluated the cell survival and engrafting at the end of the study? Moreover, given the results provided by RNAseq, it would be interesting to assess if the expression of HGF, BDNF and the other upregulated genes where also increased in the tissue area surrounding the graft, if the graft was still present.

3)      CiN cells have not been adequately studied in vitro before transplantation. Authors claim that these cells are directly committed to neuronal-like cells, thus bypassing the stage of stem cells. However, authors showed that CiN cells are immature neuronal-like cells, as they weakly express, or they don’t express at all, typical markers of mature neurons, but a characterization of stemness (e.g. nestin expression) or multipotency is lacking.

4)      The transplantation of immature CiN cells appears to produce a small functional recovery, as seen by BBB score and CatWalk analysis, but it is unclear whether this effect could be induced by an early trophic support provided by grafted cells or by functional integration of grafted cells. In order to clarify this aspect, it would be useful to assess the presence of the graft at the end of the study.

5)      The authors claim that grafted cells have the ability to promote regenerative processes within the injured spinal cord, but all data supporting this statement are not statistically significant, including MRI results, spinal cord area, amount of myelination, number of surviving motoneurons and inflammatory markers. This may be due either to a weak effect of grafted cells or to a low number of animals per group. Therefore, in my opinion, the data do not support this statement.

6)      Figure 2B shows a relative score, which is probably related to the BBB score shown in figure 2A, but a description of how this relative score has been calculated seems to be lacking in the results and in methods.

In summary, given these flaws, the manuscript could not be accepted in this form and the above-described additional information is necessary for further consideration. Moreover, conclusions must be supported by statistically significant results, and a larger sample size would be useful. In any case, a revision of data interpretation is necessary.

Author Response

We would like to express our gratitude to both the Editor and the Reviewers for the time invested in assessing our manuscript. Given the Reviewer’s positive comments, we have endeavored to address all of your concerns, and hope you will find our manuscript is now improved. We have highlighted the revised parts of the manuscript with yellow highlights.

Please find our point-by-point answers in the attached file; our answers follow your comments.

We wish to thank the reviewers again for your constructive remarks and hope that now you will find the manuscript suitable for publication.

Yours sincerely,

Ping Dai

Department of Cellular Regenerative Medicine,

Graduate School of Medical Science,

Kyoto Prefectural University of Medicine, Japan

Reviewer 2 Report

Spinal cord injury is a severe motor and sensory dysfunction frequently resulting in permanent muscle paralysis, sensory loss, and autonomic dysfunction. SCI impairments are largely attributable to the limited ability of injured neurons to regrow axons and reestablish functional connections. In the present study, the author used stem cell therapy to improve the structural integrity of the spinal cord. However, a few matters still need to address.

Q 1- Previous studies indicate NSCs/ NPCs transplantation has a beneficial effect that carries tumorigenesis possibility. Using the chemical induction method by authors is an exciting approach. However, γ-secretase inhibitors (GSIs) inhibit Notch signaling and suppress the cell proliferation of NS/PCs efficiently differentiated into neurons with limited cell proliferation after the transplantation, which may provide a better therapeutic approach. What are the author’s comments on that?

Q 2- Authors claim that CiN cells did not differentiate into neuronal-like cells eight days after the treatment. What’s the more extended time author start seeing mature neuronal marker in culture.   

Q 3- Authors used only Tuj1 as an immature neuronal maker. They need to use DCX and NeuroD1 to ensure these immature neuronal populations. 

Q 4- After the Implantation of immature CiN, how many cells survive in this process? The author needs to do a TUNEL assay. 

Q 5- BBB score at six weeks is significant between the groups. However, if we look at seven and eight, which start decreasing in CiN groups, again pointing out the survival of these cells, what authors comment on this?

Q 6- After implantation, they become a what type of cells (inhibitory or excitatory)

Q 7- Authors observed a slightly higher number of ChAT-positive cells in the immature CiN group. The author needs to colocalize the ChAT with STEM121 to make sure because of the CiN ChAT number increase. 

Q 8- Authors did RNA-seq on control and immature CiN, and 1083 genes were upregulated, and 1256 genes were downregulated in CiN cells. However, this does not signify what’s changes going on in-vivo. Authors need to do RNA-seq on spinal tissue samples or use a few selected genes as an in-situ hybridization.

Author Response

(The authors gave the same response as above.)

Round 2

Reviewer 1 Report

A gentle but inadequate answer to my first round of evaluation has been provided by authors. In particular, almost none of the issues has been addressed and the authors simply confirmed my view and, in most cases, they postponed the solution of these serious flaws to future studies. In this condition, I have no choice but to recommend rejection.

Reviewer 2 Report

The author answered all my questions. 
